# Predicting future hospital antimicrobial resistance prevalence using machine learning
Karina-Doris Vihta [1,2,3] ✉, Emma Pritchard[1,2], Koen B. Pouwels [2,4], Susan Hopkins[5], Rebecca L. Guy [5], Katherine Henderson [5], Dimple Chudasama[5], Russell Hope [5], Berit Muller-Pebody[5], Ann Sarah Walker[1,2,6,10], David Clifton [3,7,10] & David W. Eyre [1,2,6,8,9,10]

## Abstract

**Background** Predicting antimicrobial resistance (AMR), a top global health threat, nationwide at an aggregate hospital level could help target interventions. Using machine learning, we exploit historical AMR and antimicrobial usage to predict future AMR.
**Methods** Antimicrobial use and AMR prevalence in bloodstream infections in hospitals in England were obtained per hospital group (Trust) and financial year (FY, April–March) for 22 pathogen–antibiotic combinations (FY2016-2017 to FY2021-2022). Extreme Gradient Boosting (XGBoost) model predictions were compared to the previous value taken forwards, the difference between the previous two years taken forwards and linear trend forecasting (LTF). XGBoost feature importances were calculated to aid interpretability.
**Results** Here we show that XGBoost models achieve the best predictive performance. Relatively limited year-to-year variability in AMR prevalence within Trust–pathogen–antibiotic combinations means previous value taken forwards also achieves a low mean absolute error (MAE), similar to or slightly higher than XGBoost. Using the difference between the previous two years taken forward or LTF performs consistently worse. XGBoost considerably outperforms all other methods in Trusts with a larger change in AMR prevalence from FY2020-2021 (last training year) to FY2021-2022 (held-out test set). Feature importance values indicate that besides historical resistance to the same pathogen–antibiotic combination as the outcome, complex relationships between resistance in different pathogens to the same antibiotic/antibiotic class and usage are exploited for predictions. These are generally among the top ten features ranked according to their mean absolute SHAP values.
**Conclusions** Year-to-year resistance has generally changed little within Trust–pathogen–antibiotic combinations. In those with larger changes, XGBoost models can improve predictions, enabling informed decisions, efficient resource allocation, and targeted interventions.

## Plain language summary

Antibiotics play an important role in treating serious bacterial infections. However, with the increased usage of antibiotics, they are becoming less effective. In our study, we use machine learning to learn from past antibiotic resistance and usage in order to predict what resistance will look like in the future. Different hospitals across England have very different resistance levels, however, within each hospital, these levels remain stable over time. When larger changes in resistance occurred over time in individual hospitals, our methods were able to predict these. Understanding how much resistance there is in hospital populations, and what may occur in the future can help determine where resources and interventions should be directed.

Antimicrobial resistance is one of the top global health threats[1]. Bloodstream infections are typically one of the most serious types of infection; given their high mortality/morbidity, they are generally treated in hospitals and therefore are often used for surveillance of resistance. In high-income countries, any isolated pathogens will be tested for antimicrobial susceptibility against key antibiotics, while, unfortunately, most low and middle-income countries lack the laboratory capacity to test all bloodstream pathogens, if any[2]. Being able to predict future antimicrobial resistance of bloodstream infections in networks of hospitals could help target interventions and allocate resources to those most at risk, with larger predicted resistance increases or absolute rates. While estimating associations between characteristics such as age, sex and probability of resistance is important at

an individual level, it is not clear how a hospital should use this information. It may be simpler for hospitals to assume that their underlying populations are broadly similar from year to year and estimate resistance at an aggregate level. Further, empiric treatment recommendations are generally made across an entire hospital.

Antibiotic usage is a well-known driver of antibiotic resistance[3]. Several studies have investigated associations, for example using Spearman's correlation coefficients between outpatient antibiotic usage and resistance in European countries, showing countries with higher usage had higher resistance percentages[4], or using multivariate transfer functions to demonstrate positive associations between antibiotic use and resistance rates in *Pseudomonas aeruginosa* in a German hospital. The latter also allowed a decrease in resistance following usage restriction to be identified[5]. Studies have generally shown increases in usage associated with quite rapid increases in resistance, while decreases in usage were associated with no changes or very delayed and more subtle decreases[6,7]. However, exceptions have also been observed, such as increased nitrofurantoin usage leading to no changes in nitrofurantoin resistance in *Escherichia coli* urinary tract infections, while being associated with decreased trimethoprim resistance[8]. The rarity of nitrofurantoin resistance has been explained genetically by the magnitude of the distance between two genes which require inactivation[9]. A recent study used distributed lag models to estimate the relationship between relative antibiotic usage (classified as a Z-score) and antibiotic resistance at a national and international level, using 11 years of data from 26 European countries. They showed that increases in antibiotic usage Z-score were associated with an immediate and persistent increase in resistant bacteria for the four following years, while decreases in usage Z-score had little impact on resistance on the same time scale; antibiotic usage of neighbouring countries also affected resistance levels[10]. To our knowledge, whilst machine learning methods have been used in the past for predicting resistance at an individual level, for example[11,12], they have not been widely used for predicting resistance at an aggregate level such as a hospital, a network of hospitals, or a country. One study considered a feed-forward neural network with a single hidden layer, with each input neuron being a lagged time series[13]; however, while this allows for nonlinearity, it models only one time series at a time[14].

In England, National Health Service hospitals are grouped into Trusts, which are organisational units serving a geographical area or a specific speciality, therefore with multiple Trusts able to serve the same geographical area. Trusts have different antibiotic usage policies and have different resistance patterns in the population they serve. Most studies so far have focused on individual pathogens, and on understanding specifically associations between antibiotic use and antibiotic resistance in individual pathogens, with some, but not all, identifying such associations. Here, we shift focus and predict future resistance at a Trust level, exploiting all historical aggregate information that we have on each specific Trust, namely historical antibiotic usage for a variety of antibiotics, and historical antibiotic resistance to the pathogen–antibiotic of interest in bloodstream infections, but also resistance in other pathogen–antibiotic combinations and the complexity of these relationships. We explore whether a well-understood and typically successful machine learning model, namely Extreme Gradient Boosting (XGBoost), can outperform base comparators such as carrying the last value forwards, carrying the difference between the previous two years forwards and linear trend forecasting (LTF). The hypothesis is that this type of model can exploit interactions such as decreasing the use of one antibiotic leading to increasing the use of another as patients still need to be treated (e.g. ciprofloxacin use declined as it was selected for *Clostridium difficile*, and consequently amoxicillin/clavulanic acid usage increased[15]), as well as sharing of resistance mechanisms between different antibiotics and pathogens[16–18].

Here we show that XGBoost models successfully exploit information on both historical resistance prevalence in the pathogen–antibiotic of interest, but also on other pathogen–antibiotic combinations, as well as usage. Through this, they achieve better predictive performance in those Trusts where larger changes occur, without compromising forecasting capacity in those with little or no change from year to year.

## Methods
### Datasets
National antibiotic resistance data was obtained from the UK Health Security Agency's (UKHSA) Second Generation Surveillance System (SGSS), containing laboratory data supplied electronically by approximately 98% of hospital microbiology laboratories in England. As part of routine surveillance activities laboratory surveillance data were deterministically linked by UKHSA using unique patient identifiers and specimen date of collection to the Healthcare-associated Infections Data Capture System mandatory surveillance data to obtain the hospital group (Trust) for each bacteraemia case[19]. UKHSA have approval under Regulation 3 of the Health Service (Control of Patient Information) Regulations 2002 to process patient identifiable information without consent. This process considers the ethics and purpose of collecting and analysing the data, and as such ethical approval was not separately sought for this work. Aggregate data produced as part of routine surveillance activities were used for this analysis. Since all patient data was de-identified and aggregated, ethical approval for the use of data and patient consent was not required for this study. We studied pathogens isolated from bloodstream infections subject to mandatory surveillance aggregated at the Trust level (from different calendar dates, see below), and specific pathogen–antibiotic combinations, namely: methicillin-susceptible coagulase-positive *Staphylococcus* species (MSSA) (Apr2016–Mar2021): doxycycline/tetracycline, erythromycin, clarithromycin, clindamycin, vancomycin, *E. coli* (Apr2016–Mar2021) and *Klebsiella* species (Apr2017–Mar2021): ciprofloxacin, third-generation cephalosporins (resistance to any of cefotaxime, ceftazidime, cefpodoxime and ceftriaxone), gentamicin, carbapenems (either meropenem or imipenem; or ertapenem where meropenem and imipenem not tested), co-amoxiclav, piperacillin/tazobactam, *P. aeruginosa* (Apr2017–Mar2021): ciprofloxacin, ceftazidime, gentamicin, carbapenems and piperacillin/tazobactam. Aggregated monthly totals for pathogen–antibiotic combinations by Trust are only available from April 2017 for *Klebsiella* spp. and *P. aeruginosa*, and April 2016 for *E. coli* and *Staphylococcus aureus*, as the Trust assignation is obtained through linkage to mandatory surveillance data. The data quality of Trust in the laboratory reported data alone has historically been poor, and while it is improving, it is not considered sufficiently accurate for prediction. The mandatory surveillance data collection covers the *S. aureus* complex which includes *S. argenteus* and *S. Schweitzer*, although *S. aureus* is the predominant species within the MSSA and methicillin-resistant Coagulase-positive *Staphylococcus* species (MRSA) data collection. MRSA was not considered in this study as numbers per Trust are very small even when aggregating the data to financial years, with only 694 cases in 2020/2021 being reported in England[20].

Percentages of isolates with resistance to each antibiotic were calculated per Trust per financial year (FY, i.e. April–March), to keep winter months together in the same year. There was no missing data, in that every Trust had a number of pathogens tested and resistant in each year, even if both were zero. Small numbers of isolates/month for some Trusts and key pathogen–antibiotic combinations meant monthly data had to be aggregated over years to avoid large fluctuations. Isolates tested are assumed representative of bloodstream infections in each Trust, as most detectable bloodstream infections are likely ascertained by widespread testing in serious illness in a high-income setting. For some antibiotics susceptibility results were reported as susceptible or resistant. However, for a subset of the antibiotics, susceptibility results were split into susceptible, resistant, and a third intermediate category. Where susceptibility was reported as intermediate, this was considered susceptible in models following recommendations that this is susceptible under increased exposure[21].

From UKHSA, we also obtained data collected by IQVIA on monthly antibiotic usage (drug, quantity, concentration) (pharmacy dispensing) at a Trust level from April 2014[22], and used defined daily doses (DDDs)[23] per antibiotic per Trust per FY to align different drugs/concentrations. We standardised antibiotic consumption to account for Trust size[24] using Trust bed occupancy data (mean daily number of day and overnight occupied beds)[25]. Specifically, usage rates were calculated as the total DDDs per month divided by the number of days in the month, to obtain the mean DDDs

per day, divided by the mean number of day and overnight occupied beds per day and multiplied by 100. For example, an antibiotic usage of 40 DDDs amoxicillin per 100 bed-days, that is a usage rate of 40%, means 40% of inpatients receive one DDD of amoxicillin every day, an estimate of the therapeutic intensity. Trust mergers were carried backwards in time, such that results are presented based on Trusts existing as distinct entities in 2021[26]. In the commonly used antibiotics, data was available for all Trusts across all FYs, with very few exceptions, namely, one trust missing data usage across all antibiotics in 2019-2020 and a further two in 2020–2021 (Supplementary Data 1). In the less commonly used antibiotics, missing data was very common, although this may indicate zero usage for those years. In our models we only included the top 34 antibiotics (based on mean usage rate across all Trust-FYs >1%) plus ertapenem (mean usage rate just below 1%, but an antibiotic of interest as it is a carbapenem, the broadest spectrum antibiotic class currently in reasonably wide usage).

## XGBoost models
Our main goal was to predict future antibiotic resistance for each Trust and pathogen–antibiotic combination based on historical resistance and antibiotic usage. We fit separate models for each pathogen–antibiotic combination as the outcome, but included prior antibiotic consumption data for all antibiotics and prior resistance data for all pathogen–antibiotic combinations in each model. We also explored the predictive performance of historical usage alone. Hence, each Trust contributed a training example to each model, containing information on all available prior annual usage and pathogen–antibiotic resistance prevalences. We explored whether a previously highly successful machine learning model with the ability to learn non-linear relationships and interactions between different features, namely XGBoost[27], could outperform base comparators. XGBoost is not designed for time series, but with appropriate feature engineering and setup can be used for time series forecasting, especially as our time series are very short. Specifically, we used a training-test data split based on calendar time to train models and evaluate performance. We used percentage resistance in FY2020–2021 as our outcome for our training dataset. Although this FY includes the start of the COVID-19 pandemic, this was the closest to the test set (outcome FY2021–2022), and maximised the history available for model training. All data available from prior years of the time series was provided as input. Each pathogen–antibiotic-FY resistance prevalence and each antibiotic-FY usage rate with available data contributed a feature to the input matrix, with each Trust contributing data for one row. FY2021-2022 was used as the outcome for our test set. We excluded Trusts testing <100 isolates per year throughout the period studied and Trust–pathogen–antibiotic-FYs with ≤10 susceptibility results to avoid fluctuations due to small numbers unduly influencing results (arbitrary thresholds). Six FYs of historical antibiotic usage were available for training (from April 2014), four FYs of historical resistance (from April 2016) for *E. coli* and MSSA, and three FYs (from April 2017) of data respectively for both *Klebsiella* sp. and *P. aeruginosa*. When exploring predictive performance with 3, 2 and 1 FY(s) historical data, we increased the size of the training dataset by considering previous years as additional outcomes. As the test dataset remained unchanged, predictive performance results were comparable. XGBoost models were fitted with both default and tuned hyperparameters. To improve generalisability, 3-fold cross-validation on the training dataset was used to tune model hyperparameters, i.e., the number of estimators, the maximum depth and the minimum child weight (see Supplementary Information for full details). For XGBoost models using historical antibiotic usage alone (no information on previous resistance prevalence) only models with default parameters were fitted (see Supplementary Information for full details). We also explored whether re-fitting models choosing only features with feature importance above white noise improved performance. Feature importance was captured using mean absolute SHapley Additive exPlanations (SHAP)[28] computed on the training dataset.

## Performance evaluation
We chose to minimise the mean absolute error (mean of the absolute difference between true and predicted value) as it is easily interpretable

and less influenced by outliers than root mean squared error, as for the former all errors are given the same weight, while for the latter more weight is given to larger errors. We wished to avoid over-influence from outliers as despite data cleaning, it is quite likely that some large outliers still arise from data quality issues. We are mostly concerned with optimising performance for most Trusts rather than those with large errors and hence choose to optimise mean absolute error. XGBoost handles missing values by default, by learning at a split of a decision tree which classification of the missing value group into each split minimises the mean absolute error, and making that classification. Missing data was present in resistance because of excluding Trust–pathogen–antibiotic-FYs with <10 results (Supplementary Tables 1 and 2), and in usage for only a few Trusts in the highest usage antibiotics (Supplementary Data 1).

## Model interpretability
To aid model interpretability, global feature importance was captured through mean absolute SHAP values calculated across all observations. SHAP values measure the impact each feature has on the individual predictions. These were computed on the test set for each Trust in each individual pathogen–antibiotic combination model. Large positive/negative values indicate features that have a considerable impact on the prediction. Therefore, higher mean absolute SHAP values (calculated across all Trusts) indicate more influential features overall.

## Base comparators
We compared XGBoost with carrying the last value forwards, carrying the difference between the previous two years forwards and LTF which only considers one time series from a single Trust at a time (compared to XGBoost models which consider data across all Trusts, as well as multiple time series for each Trust as features). When comparing performance between different models, if, for example, the previous value was missing and therefore a prediction could not be made for this model, these Trust–pathogen–antibiotic-FYs were dropped and mean absolute errors were calculated only in Trust–pathogen–antibiotic-FYs for which predictions could be made for all models being compared.

## Statistics and reproducibility
We summarised data characteristics and machine learning model performance using percentages. No statistical tests were conducted and no statistical regression models were fitted.

## Reporting summary
Further information on research design is available in the Nature Portfolio Reporting Summary linked to this article.

## Results
### Data summary
Susceptibility data were available for 138 hospital groups (Trusts) for FYs between April 2016 and March 2022 for *E. coli* and MSSA, and April 2017 and March 2022 for *Klebsiella* sp. and *P. aeruginosa*. 19 Trusts with a maximum <100 tested isolates/FY across all pathogen–antibiotic combinations were excluded completely, as small sample sizes made resistance percentages highly variable from year to year (Supplementary Table 3). 16/19 excluded Trusts were specialist Trusts with typically much lower rates of bloodstream infection. Trust–pathogen–antibiotic-FYs with ≤10 susceptibility results were also excluded (Supplementary Tables 1 and 2) for similar reasons (Supplementary Fig. 1).

Antibiotic resistance prevalence varied by pathogen, antibiotic, and between Trusts over the study period (Fig. 1a). For example, within *E. coli* the median overall resistance prevalence for amoxicillin/clavulanic acid was 43%, vs 9% for piperacillin/tazobactam, but with wide interquartile ranges (IQR) (36–49% and 6–12% respectively), reflecting Trust-level variation. However, there was much less variability within each Trust over time for a given pathogen–antibiotic combination, with >75% of Trusts having a standard deviation (across annual resistance prevalences) of <8% even for those

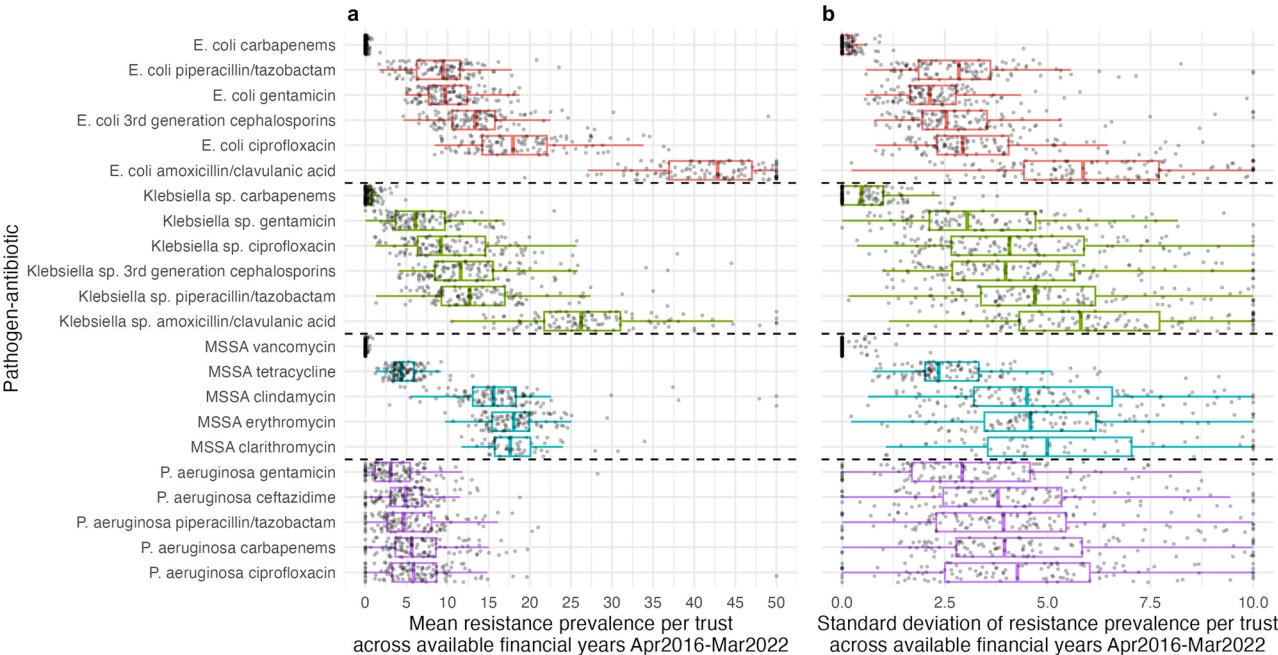

**Fig. 1 | Mean resistance prevalence and standard deviation per Trust–pathogen–antibiotic.** Distribution of mean resistance prevalence (**a**) and standard deviation (**b**) per Trust–pathogen–antibiotic across available financial years (Apr2016–Mar2022 for *E. coli* and MSSA and Apr2017–Mar2022 for *Klebsiella* sp. and *P. aeruginosa*). *n* = 119 Trusts, however not all Trusts contributed data to each boxplot (Supplementary Table 2). Red (*E. coli*), green (*Klebsiella* sp.), blue (MSSA), purple (*P. aeruginosa*). Note: one point per Trust. Outliers outside of the *x*-axis scale (>50 left panel >10 right panel) were truncated. Centre line, median; box limits, upper and lower quartiles; whiskers, 1.5× interquartile range.

pathogen–antibiotic combinations with the highest standard deviations (*E. coli*-amoxicillin/clavulanic acid and *Klebsiella* sp.-amoxicillin/clavulanic acid, Fig. 1b). We observed uncommon outliers which may indicate data quality issues; these were not excluded from analyses as they could also represent outbreaks. Distributions of antibiotic resistance within a pathogen–antibiotic combination were broadly similar across the FYs (Supplementary Fig. 2).

### FY-to-FY changes in resistance prevalence
Overall Trust-FYs, the median difference between current and previous resistance prevalence within each pathogen–antibiotic combination was always within ±1%; 18/22 pathogen–antibiotic combinations had a median within ±0.5% (Fig. 2a). Considering individual years, 95% of Trust–pathogen–antibiotic-FYs differed in the resistance prevalence compared with the previous year by <10% and 84% <5%. The largest absolute differences were observed for amoxicillin/clavulanic acid resistance in *Klebsiella* sp., but even there 43% of Trust-FYs had absolute differences <5%. Distributions and percentages were broadly similar over time (Figs. S3 and S4). The median LTF estimated change between FYs 2016–2017 and 2021–2022 was <2.5% for 18/22 pathogen–antibiotic combinations and <5% for the remaining 4; 82% of Trust–pathogen–antibiotic-FYs combinations had an LTF-estimated absolute change across the 6 FYs <10%, and 60% <5% (Fig. 2b).

### FY-to-FY changes in usage
Antibiotic usage rates were available for all 119 Trusts from 2014–2015 to 2020–2021. Similarly to resistance prevalences, antibiotic usage rates varied between the different antibiotics (Supplementary Fig. 5), but with relatively little change over time for many antibiotics (Supplementary Fig. 6). Of the most commonly used antibiotics, there was a small increase in amoxicillin/clavulanic acid usage (median across Trusts 24% (IQR 15–33%) in FY2014–2015 to 32% (22–42%) in FY2020–2021), a decrease in trimethoprim usage (median 8% (5–10%) to 3% (2–5%) respectively), and a corresponding increase in sulfamethoxazole/trimethoprim (median 3% (2–5%) to 6% (4–9%), respectively) and in nitrofurantoin (median 2% (2–4%) to 5%

(4–7%), respectively) (reflecting change in antibiotic recommendations for treating urinary tract infections). There was a decrease in piperacillin/tazobactam usage in FY2017–2018 to median 3% (2–4%) (vs 5% (4–7%) in FY2014–2015), resulting from shortages due to an explosion at a Chinese antibiotics factory[29], followed by a slow rise back to similar levels by FY2020–2021 (median 5% (3–7%)). There was very little difference from year to year within a Trust, except for a few outliers, that may indicate potential data quality issues rather than true changes, potentially excepting supply interruptions and/or COVID-19 impacts (Figs S7 and S8).

### Forecasting performance
Overall, XGBoost models achieved the best predictive performance, with previous value taken forwards having very similar or slightly higher mean absolute error across all Trusts (Fig. 3). The largest differences between previous value taken forwards and XGBoost were when XGBoost outperformed previous value taken forwards, eg for *P. aeruginosa* ceftazidime (2% difference, from 4% to 6%). The mean absolute error from using the previous resistance prevalence taken forwards by pathogen–antibiotic-FY over the Trusts was similar over time (Supplementary Fig. 9) and approximately proportional to the mean resistance level across Trusts (Fig. 3). Taking the difference between the previous two years forwards performed the worst across all pathogen–antibiotic combinations, having the highest mean absolute error, followed by LTF. For three pathogen–antibiotic combinations, carbapenem resistance in *E. coli* and *Klebsiella* sp. and vancomycin resistance in MSSA, most Trusts had 0% resistance for all available FYs (Supplementary Table 4). This was reflected in the considerably lower mean absolute error. XGBoost models with 3, 2 and 1 FY(s) historical usage and resistance data increased the size of the training dataset by considering previous years as additional outcomes. These had very similar performance (Supplementary Fig. 10). Differences in performance between XGBoost with and without feature selection were very small and neither outperformed the other across all pathogen–antibiotic combinations (Supplementary Fig. 11).

Focusing on evaluating performance in those Trusts where there was the biggest absolute difference between the resistance prevalence in FYs

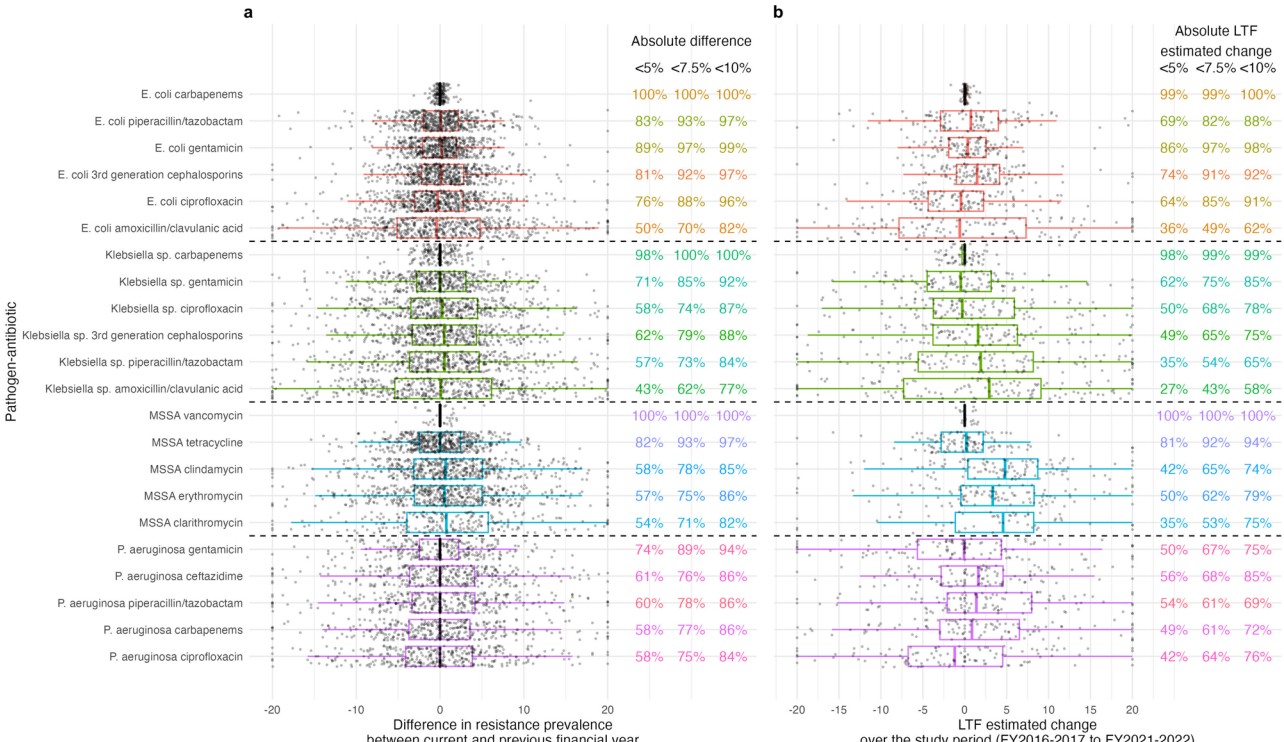

**Fig. 2 | Difference in resistance prevalence between the current and previous financial year and estimated change over the study period.** Distribution of difference in resistance prevalence between the current and previous financial year (**a**) and LTF estimated change over the study period (**b**), per pathogen–antibiotic combination across all Trusts and financial years. Percentages of Trust-FYs that have an absolute difference <5%, <7.5% and <10% between the current and the previous financial year are also given by pathogen–antibiotic combination (**a**), and an absolute LTF estimated change <5%, <7.5% and <10% (**b**). *n* = 119 Trusts × 5 FY-to-FY differences (*E. coli* and MSSA) and 4

FY-to-FY differences (*Klebsiella* sp. and *P. aeruginosa*), however not all Trusts contributed data to each boxplot (Supplementary Table 2). Red (*E. coli*), green (*Klebsiella* sp.), blue (MSSA), purple (*P. aeruginosa*). Note: one point per trust year. Distribution split by financial year available in Supplementary Fig. 3. Percentage of trusts with an absolute difference in resistance prevalence <5%, <7.5%, and <10% split by financial year in Supplementary Fig. 4. Outliers outside of *x*-axis scale (absolute value > 20) were truncated. Centre line, median; box limits, upper and lower quartiles; whiskers, 1.5× interquartile range.

2021–2022 and 2020–2021, considering an arbitrary threshold of >10% (Fig. 4), XGBoost outperformed previous value taken forwards in all but one pathogen–antibiotic combination (*E. coli*–gentamicin). Performance gains were substantially larger in magnitude in this subgroup, while there was little to no difference in the mean absolute error in the remaining Trusts (≤10% difference). Results were similar for thresholds for the difference between resistance prevalence of 7.5% and 5%, where the outperformance by XGBoost occurred across all pathogen–antibiotic combinations including *E. coli*–gentamicin (Supplementary Fig. 12). In Trusts where the absolute difference was >10%, there were both increases and decreases from the previous resistance prevalence in 17/22 pathogen–antibiotic combinations. Performance gains in mean absolute error were observed both in Trusts with positive and negative differences between current and previous resistance prevalences (Supplementary Fig. 13).

**Model interpretability**
Considering model interpretability, generally previous resistance prevalence to the same pathogen–antibiotic combination as the outcome was among the top 10 features ranked according to their mean absolute SHAP values (Tables 1–4). Previous resistance prevalence to the same antibiotic but in a different pathogen, as well as usage of the same antibiotic, were also generally among the top ten features, and similarly for other antibiotics from the same class.

**Discussion**
While associations between antibiotic usage and antibiotic resistance are widely accepted, here we have built a model that allows us to take advantage

of the complex relationship between the usage of different antibiotics, and between different resistance mechanisms being responsible for resistance to multiple antibiotics or in multiple pathogens, with the goal of predicting future resistance at an aggregate level for a hospital group. Features with the highest contributions to the prediction illustrated that such complex relationships were very likely captured and exploited by the models. One key challenge is that changes in resistance were small for many pathogen–antibiotic combinations we considered. Training the model on all Trusts, but evaluating performance in the subgroup of Trusts where changes from one financial year to the next were the largest, we achieved better predictive performance when considering the mean absolute error, without compromising predictive performance in those where the changes were minimal.

Relatively few studies have considered multiple pathogen–antibiotics simultaneously. One previous study considered forecasting quarterly resistance in *E. coli* bloodstream infections to third-generation cephalosporins, ciprofloxacin, gentamicin, and piperacillin/tazobactam per clinical commissioning group (CCG, groups of general practices) in England using data from October 2015 to October 2018, as well as annual resistance in European countries to carbapenems and fluoroquinolones in *K. pneumoniae*, *E. coli*, *P. aeruginosa*, and *Acinetobacter* spp. using data from 2012 to 2016[14]. They compared the last value taken forwards with single time series models allowing for more complexity including autoregressive integrated moving average (ARIMA), Expected–Trend–Seasonal, and a feed-forward neural network with a single hidden layer, with each input neuron being a lagged time series, as well as fitting an integrated nested Laplace approximations

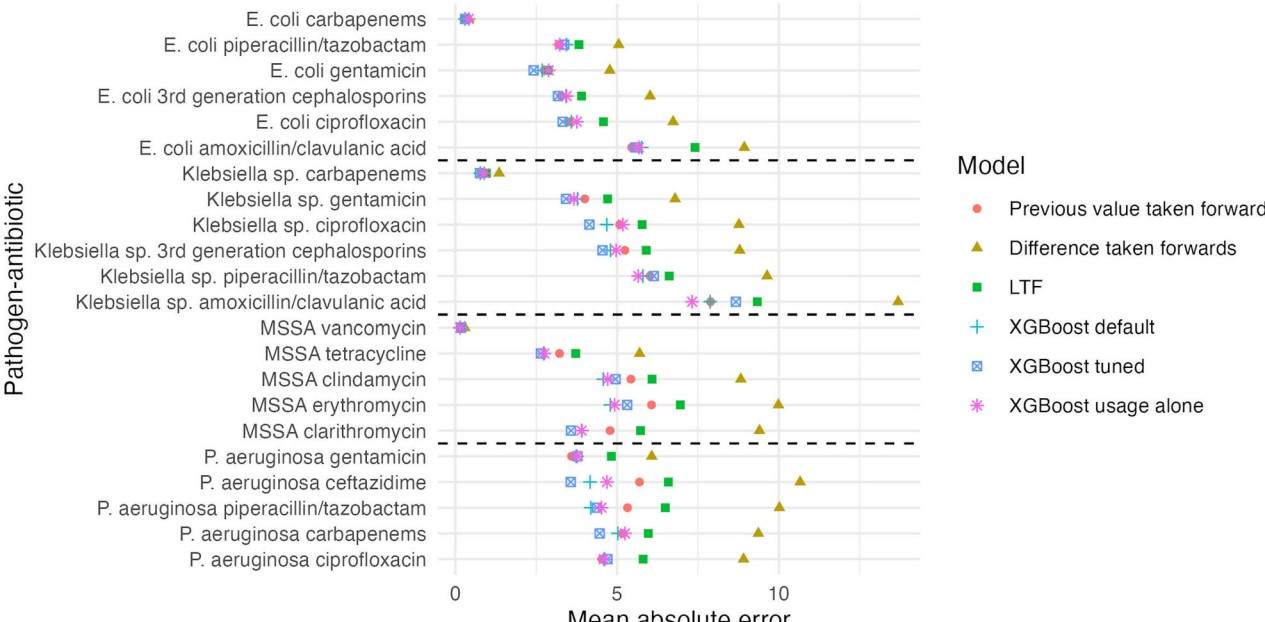

**Fig. 3 | Predictive performance comparison.** Mean absolute error for prediction on test set (resistance prevalence in FY2021-2022) for 6 different prediction models: taking the previous value forwards (red circle), taking the difference forwards (yellow triangle), LTF (green square), XGboost with default parameters (light blue plus sign), XGboost with tuned hyperparameters (dark blue square with multiplication sign), and XGboost with previous antibiotic usage alone as input features

(with default parameters, no information on previous resistance prevalence, pink star). *n* = 119 Trusts, however not all Trusts contributed data to each boxplot (Supplementary Table 2). Note: 70 residuals that had either missing previous values or previous differences were excluded for comparability of performance measures between the models, although XGboost also made these predictions.

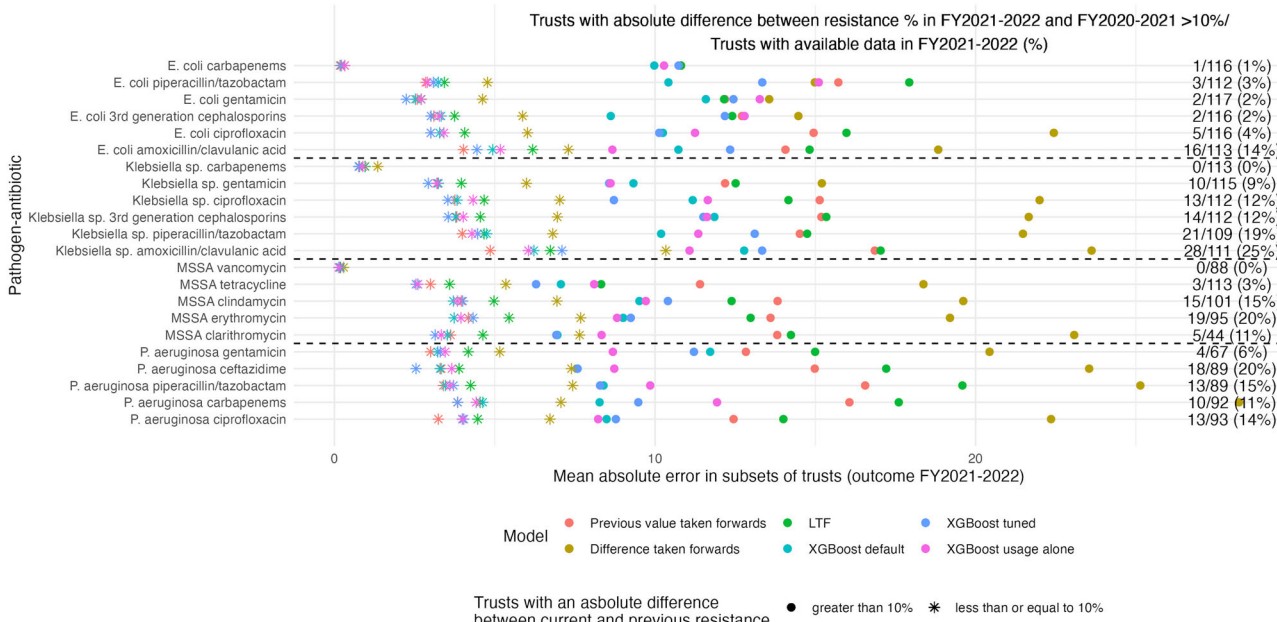

**Fig. 4 | Predictive performance comparison split by absolute difference between consecutive years.** Mean absolute error for prediction on the test set (resistance prevalence in FY2021–2022) for six different prediction models split by the absolute difference between FY2021–2022 and FY2020–2021 in resistance prevalence, >10% (full circle) or ≤10% (star). Six different prediction models: taking the previous value forwards (red), taking the difference forwards (yellow), LTF (green), XGboost with default parameters (light blue), XGboost with tuned hyperparameters (dark blue), and XGboost with previous antibiotic usage alone as input features (with default

parameters, no information on previous resistance prevalence, pink). *n* Trusts included are provided in the Figure. Note: 70 residuals that had either missing previous values or previous differences were excluded for comparability of performance measures between the models, although XGboost also made these predictions. Results using thresholds of 7.5% and 5% are illustrated in Supplementary Fig. 11. For 3 pathogen–antibiotic combinations: *E. coli* carbapenems, *Klebsiella* sp. carbapenems and MSSA vancomycin, most Trusts had 0% resistance prevalence for all available FYs (Supplementary Table 2).

## Table 1 | *E. coli* resistance prevalence: model interpretability

**E. coli resistance**

| Feature ranking by mean absolute SHAP values over Trusts | Amoxicillin/clavulanic acid | Carbapenems | Third generation cephalosporins | Ciprofloxacin | Gentamicin | Piperacillin/tazobactam |
|---|---|---|---|---|---|---|
| 1 | **E. coli amoxicillin/clavulanic acid-res, prev1 2.87** | Usage-colistin, prev1, 0.06 | E. coli gentamicin-res, prev2, 1.01 | **E. coli ciprofloxacin-res, prev1, 1.57** | **E. coli gentamicin-res, prev1, 0.55** | **E. coli piperacillin/tazobactam-res, prev1, 2.1** |
| 2 | Usage-cefalexin, prev6 0.87 | Usage-pivmecillinam, prev3, 0.06 | *Klebsiella sp. amoxicillin/clavulanic acid-res, prev2, 0.69* | **E. coli ciprofloxacin-res, prev2, 1.47** | **E. coli gentamicin-res, prev2, 0.45** | *Klebsiella sp. gentamicin-res, prev3, 0.72* |
| 3 | E. coli gentamicin-res, prev1 0.38 | Usage-flucloxacillin, prev5, 0.06 | *Klebsiella sp. 3rd generation cephalosporins-res, prev2, 0.61* | MSSA clindamycin-res, prev2, 0.49 | **E. coli gentamicin-res, prev3, 0.44** | Usage-penicillin v, prev1, 0.46 |
| 4 | Usage-cefuroxime, prev6 0.36 | **E. coli carbapenems-res, prev3, 0.04** | **E. coli 3rd generation cephalosporins-res, prev2, 0.6** | E. coli 3rd generation cephalosporins-res, prev2, 0.48 | *Klebsiella sp. gentamicin-res, prev1, 0.32* | Usage-meropenem, prev3, 0.42 |
| 5 | *P. aeruginosa gentamicin-res, prev1 0.35* | *Klebsiella sp. carbapenems-res, prev1, 0.04* | Usage-amoxicillin, prev1, 0.42 | Usage-cefotaxime, prev4, 0.4 | E. coli ciprofloxacin-res, prev3, 0.29 | *Klebsiella sp. piperacillin/tazobactam-res, prev3, 0.31* |
| 6 | Usage-cefuroxime, prev3 0.3 | MSSA clindamycin-res, prev4, 0.04 | Usage-penicillin g, prev5, 0.37 | **E. coli ciprofloxacin-res, prev3, 0.32** | Usage-tobramycin, prev2, 0.29 | *P. aeruginosa ceftazidime-res, prev1, 0.22* |
| 7 | *P. aeruginosa ciprofloxacin-res, prev1 0.29* | Usage-azithromycin, prev6, 0.02 | E. coli gentamicin-res, prev3, 0.33 | *Klebsiella sp. gentamicin-res, prev1, 0.31* | Usage-lymecycline, prev5, 0.26 | Usage-penicillin v, prev5, 0.2 |
| 8 | Usage-cefuroxime, prev1 0.29 | *Klebsiella sp. 3rd generation cephalosporins-res, prev1, 0.02* | E. coli ciprofloxacin-res, prev3, 0.33 | Usage-moxifloxacin, prev5, 0.27 | *Klebsiella sp. piperacillin/tazobactam-res, prev2, 0.25* | Usage-moxifloxacin, prev5, 0.18 |
| 9 | MSSA clindamycin-res, prev1 0.28 | MSSA clindamycin-res, prev1, 0.02 | *Klebsiella sp. 3rd generation cephalosporins-res, prev3, 0.32* | Usage-penicillin v, prev6, 0.26 | Usage-cefuroxime, prev1, 0.21 | Usage-tobramycin, prev1, 0.18 |
| 10 | **E. coli amoxicillin/clavulanic acid-res, prev4 0.26** | E. coli gentamicin-res, prev1, 0.02 | E. coli ciprofloxacin-res, prev4, 0.28 | *Klebsiella sp. carbapenems-res, prev1, 0.26* | Usage-linezolid, prev5, 0.2 | Usage-meropenem, prev1, 0.17 |
| Legend | **Same pathogen–antibiotic** | *Same antibiotic* | Same class antibiotic | res = resistance | prevX = X year(s) in the past | |

The top ten features ranked according to mean absolute SHAP values for *E. coli* resistance prevalence prediction models (one per antibiotic), calculated on the test dataset.

Note: **same pathogen–antibiotic** shown in **bold**, *same antibiotic* shown in *italics*, same class antibiotics underlined.

**Table 2 | *Klebsiella* sp. resistance prevalence: model interpretability**

***Klebsiella* sp. resistance**

| Feature ranking by mean absolute SHAP values over trusts | Amoxicillin/clavulanic acid | Carbapenems | Third generation cephalosporins | Ciprofloxacin | Gentamicin | Piperacillin/ tazobactam |
|---|---|---|---|---|---|---|
| 1 | Klebsiella sp. 3rd generation cephalosporins-res, prev1, 1.48 | *E. coli carbapenems-res, prev2, 0.19* | **Klebsiella sp. 3rd generation cephalosporins-res, prev1, 1.76** | *Klebsiella sp. amoxicillin/ clavulanic acid-res, prev2, 1.03* | *Klebsiella sp. ciprofloxacin-res, prev1, 0.77* | *Klebsiella sp. piperacillin/ tazobactam-res, prev1, 1.4* |
| 2 | *Klebsiella sp. gentamicin-res, prev1, 1* | Usage-levofloxacin, prev5, 0.14 | Usage-ceftazidime, prev1, 0.73 | **Klebsiella sp. ciprofloxacin-res, prev1, 0.69** | **Klebsiella sp. gentamicin-res, prev1, 0.58** | *E. coli gentamicin-res, prev2, 0.86* |
| 3 | **Klebsiella sp. amoxicillin/ clavulanic acid-res, prev1, 0.97** | Usage-colistin, prev6, 0.11 | Usage-penicillin v, prev3, 0.6 | Klebsiella sp. 3rd generation cephalosporins-res, prev1, 0.61 | E. coli 3rd generation cephalosporins-res, prev3, 0.51 | Klebsiella sp. amoxicillin/ clavulanic acid-res, prev3, 0.82 |
| 4 | **Klebsiella sp. amoxicillin/ clavulanic acid-res, prev2, 0.82** | **Klebsiella sp. carbapenems-res, prev1, 0.1** | Klebsiella sp. ciprofloxacin-res, prev3, 0.39 | **Klebsiella sp. ciprofloxacin-res, prev3, 0.53** | MSSA erythromycin-res, prev1, 0.29 | *E. coli gentamicin-res, prev4, 0.6* |
| 5 | E. coli amoxicillin/clavulanic acid-res, prev1, 0.51 | P. aeruginosa carbapenems-res, prev2, 0.1 | E. coli piperacillin/tazobactam-res, prev1, 0.39 | Usage-ofloxacin, prev6, 0.42 | Usage-ceftriaxone, prev3, 0.28 | *Usage-piperacillin/ tazobactam, prev4, 0.46* |
| 6 | Usage-teicoplanin, prev6, 0.46 | Usage-linezolid, prev6, 0.1 | **Klebsiella sp. 3rd generation cephalosporins-res, prev2, 0.38** | Klebsiella sp. gentamicin-res, prev3, 0.39 | Usage-moxifloxacin, prev1, 0.27 | *E. coli piperacillin/ tazobactam-res, prev1, 0.45* |
| 7 | Usage-penicillin v, prev4, 0.41 | P. aeruginosa ceftazidime-res, prev2, 0.1 | P. aeruginosa ceftazidime-res, prev1, 0.31 | Usage-flucloxacillin, prev6, 0.37 | Usage-clarithromycin, prev1, 0.26 | Klebsiella sp. ciprofloxacin-res, prev3, 0.43 |
| 8 | Usage-ceftazidime, prev6, 0.37 | Klebsiella sp. carbapenems-res, prev3, 0.09 | *E. coli 3rd generation cephalosporins-res, prev1, 0.31* | Usage-ofloxacin, prev4, 0.34 | Usage-colistin, prev3, 0.25 | **Klebsiella sp. piperacillin/ tazobactam-res, prev2, 0.36** |
| 9 | Usage-lymecycline, prev6, 0.35 | E. coli carbapenems-res, prev4, 0.08 | Usage-piperacillin/tazobactam, prev4, 0.3 | Usage-linezolid, prev3, 0.32 | E. coli amoxicillin/clavulanic acid-res, prev4, 0.24 | Klebsiella sp. ciprofloxacin-res, prev2, 0.31 |
| 10 | Usage-colistin, prev1, 0.35 | MSSA clindamycin-res, prev2, 0.08 | Klebsiella sp. gentamicin-res, prev1, 0.28 | Klebsiella sp. 3rd generation cephalosporins-res, prev2, 0.29 | P. aeruginosa piperacillin/ tazobactam-res, prev2, 0.21 | Klebsiella sp. amoxicillin/ clavulanic acid-res, prev2, 0.28 |
| Legend | **Same pathogen–antibiotic** | *Same antibiotic* | Same class antibiotic | | prevX = X year(s) in the past | res = resistance |

The top ten features ranked according to mean absolute SHAP values for *Klebsiella* sp. resistance prevalence prediction models (one per antibiotic), calculated on the test dataset.
Note: **same pathogen–antibiotic** shown in **bold**, *same antibiotic* shown in *italics*, same class antibiotics underlined.

**Table 3 | MSSA resistance prevalence: model interpretability**

| MSSA resistance | | | | | |
|---|---|---|---|---|---|
| Feature ranking by mean absolute SHAP values over trusts | Clarithromycin | Clindamycin | Erythromycin | Tetracycline | Vancomycin |
| 1 | Usage-ertapenem, prev6, 0.75 | Usage-lymecycline, prev4, 1.21 | Usage-flucloxacillin, prev6, 0.77 | *E. coli* ciprofloxacin-res, prev2, 0.42 | *Klebsiella* sp. carbapenems-res, prev1, 0.08 |
| 2 | *P. aeruginosa* carbapenems-res, prev1, 0.72 | **MSSA clindamycin-res, prev1, 0.81** | *Klebsiella* sp. 3rd generation cephalosporins-res, prev1, 0.58 | Usage-sulfamethoxazole/trimethoprim, prev3, 0.4 | Usage-tobramycin, prev1, 0.07 |
| 3 | Usage-flucloxacillin, prev5, 0.71 | *P. aeruginosa* gentamicin-res, prev3, 0.62 | *Klebsiella* sp. amoxicillin/clavulanic acid-res, prev1, 0.48 | Usage-sulfamethoxazole/trimethoprim, prev1, 0.37 | *E. coli* amoxicillin/clavulanic acid-res, prev4, 0.04 |
| 4 | *E. coli* piperacillin/tazobactam-res, prev2, 0.65 | Usage-amoxicillin, prev1, 0.44 | <u>Usage-clarithromycin, prev6, 0.39</u> | Usage-ciprofloxacin, prev1, 0.29 | *Klebsiella* sp. amoxicillin/clavulanic acid-res, prev1, 0.03 |
| 5 | Usage-linezolid, prev5, 0.55 | <u>Usage-azithromycin, prev4, 0.37</u> | Usage-clarithromycin, prev1, 0.38 | *Klebsiella* sp. ciprofloxacin-res, prev2, 0.27 | Usage-tobramycin, prev3, 0.02 |
| 6 | <u>MSSA erythromycin-res, prev2, 0.49</u> | <u>MSSA clarithromycin-res, prev2, 0.34</u> | *E. coli* amoxicillin/clavulanic acid-res, prev4, 0.36 | MSSA clindamycin-res, prev3, 0.23 | *E. coli* piperacillin/tazobactam-res, prev1, 0.01 |
| 7 | Usage-colistin, prev1, 0.45 | *E. coli* piperacillin/tazobactam-res, prev1, 0.32 | *E. coli* gentamicin-res, prev3, 0.35 | *E. coli* amoxicillin/clavulanic acid-res, prev1, 0.23 | *P. aeruginosa* ciprofloxacin-res, prev2, 0.01 |
| 8 | **MSSA clarithromycin-res, prev2, 0.39** | <u>MSSA erythromycin-res, prev2, 0.31</u> | <u>MSSA clindamycin-res, prev1, 0.33</u> | *E. coli* 3rd generation cephalosporins-res, prev1, 0.18 | Usage-gentamicin, prev6, 0.01 |
| 9 | *Klebsiella* sp. amoxicillin/clavulanic acid-res, prev3, 0.33 | *E. coli* 3rd generation cephalosporins-res, prev4, 0.3 | *Klebsiella* sp. ciprofloxacin-res, prev1, 0.29 | MSSA erythromycin-res, prev2, 0.18 | *P. aeruginosa* ciprofloxacin-res, prev1, 0.01 |
| 10 | Usage-pivmecillinam, prev3, 0.33 | *Klebsiella* sp. piperacillin/tazobactam-res, prev2, 0.26 | *P. aeruginosa* ciprofloxacin-res, prev1, 0.28 | Usage-azithromycin, prev3, 0.17 | Usage-cefotaxime, prev1, 0.01 |
| Legend | **Same pathogen–antibiotic** | *Same antibiotic* | <u>Same class antibiotic</u> | res = resistance | prevX = X year(s) in the past |

The top ten features ranked according to mean absolute SHAP values for MSSA resistance prevalence prediction models (one per antibiotic), calculated on the test dataset.
Note: **same pathogen–antibiotic** shown in **bold**, *same antibiotic* shown in *italics*, <u>same class antibiotics underlined</u>.

**Table 4 | *P. aeruginosa* resistance prevalence: model interpretability**

| *P. aeruginosa* resistance | | | | | |
|---|---|---|---|---|---|
| Feature ranking by mean absolute SHAP values over trusts | Carbapenems | Ceftazidime | Ciprofloxacin | Gentamicin | Piperacillin/tazobactam |
| 1 | *Usage-meropenem, prev6, 1.07* | *P. aeruginosa ciprofloxacin-res, prev2, 0.56* | *P. aeruginosa gentamicin-res, prev1, 0.74* | ***P. aeruginosa gentamicin-res, prev1, 0.98*** | Usage-sulfamethoxazole/trimethoprim, prev2, 0.77 |
| 2 | *Usage-meropenem, prev1, 0.71* | MSSA clindamycin-res, prev4, 0.44 | *Usage-ciprofloxacin, prev4, 0.67* | Usage-erythromycin, prev6, 0.86 | MSSA clindamycin-res, prev1, 0.65 |
| 3 | Usage-ciprofloxacin, prev1, 0.71 | *P. aeruginosa ciprofloxacin-res, prev3, 0.38* | *P. aeruginosa ceftazidime-res, prev1, 0.62* | Usage-levofloxacin, prev5, 0.45 | E. coli amoxicillin/clavulanic acid-res, <u>prev1, 0.48</u> |
| 4 | <u>Usage-cefotaxime, prev2, 0.49</u> | Usage-teicoplanin, prev4, 0.31 | Usage-tobramycin, prev3, 0.48 | Usage-erythromycin, prev1, 0.28 | MSSA erythromycin-res, prev3, 0.46 |
| 5 | Usage-cefuroxime, prev4, 0.3 | <u>Klebsiella sp. 3rd generation cephalosporins-res, prev2, 0.3</u> | Usage-trimethoprim, prev3, 0.45 | *Usage-gentamicin, prev1, 0.28* | *Usage-piperacillin/tazobactam, prev4, 0.43* |
| 6 | Usage-nitrofurantoin, prev2, 0.25 | <u>Usage-cefalexin, prev1, 0.28</u> | MSSA erythromycin-res, prev1, 0.34 | Usage-colistin, prev4, 0.27 | *Klebsiella sp. piperacillin/tazobactam-res, prev3, 0.33* |
| 7 | Usage-cefotaxime, prev1, 0.24 | *P. aeruginosa carbapenems-res, <u>prev1, 0.26</u>* | Usage-clarithromycin, prev1, 0.33 | Usage-clindamycin, prev1, 0.27 | MSSA erythromycin-res, prev1, 0.3 |
| 8 | <u>Klebsiella sp. piperacillin/tazobactam-res, prev3, 0.24</u> | Usage-tobramycin, prev1, 0.26 | *P. aeruginosa ceftazidime-res, prev3, 0.31* | E. coli piperacillin/tazobactam-res, prev3, 0.24 | Usage-nitrofurantoin, prev1, 0.29 |
| 9 | Usage-sulfamethoxazole/trimethoprim, prev6, 0.22 | *Klebsiella sp. ciprofloxacin-res, prev1, 0.23* | Usage-vancomycin iv, prev5, 0.31 | Usage-ceftazidime, prev6, 0.23 | *P. aeruginosa ceftazidime-res, <u>prev1, 0.29</u>* |
| 10 | Usage-linezolid, prev6, 0.22 | MSSA clarithromycin-res, prev4, 0.23 | Usage-tobramycin, prev1, 0.3 | E. coli carbapenems-res, prev1, 0.21 | MSSA clindamycin-res, prev4, 0.29 |
| Legend | **Same pathogen–antibiotic** | *Same antibiotic* | <u>Same class antibiotic</u> | res = resistance | prevX = X year(s) in the past |

The top ten features ranked according to mean absolute SHAP values for *P. aeruginosa* resistance prevalence prediction models (one per antibiotic), calculated on the test dataset.
Note: **same pathogen–antibiotic** shown in **bold**, *same antibiotic* shown in *italics*, <u>same class antibiotics underlined</u>.

spatiotemporal model to all groups (CCGs or countries) and forecasting for each individual time series, while also including covariates such as antibiotic usage. They found that the median root mean square error across each pathogen–antibiotic combination was relatively small (range 0–7%). Similarly to our study, the last value taken forward outperformed the other predictors when considering aggregate performance measures for yearly European resistance data, despite the spatiotemporal model being able to capture and account for associations between antibiotic usage and resistance. At the CCG level, the more complex Expected–Trend–Seasonal model captured some seasonality and improved predictive performance, but only very slightly compared to the previous value taken forward. Traditional time series generally consider modelling one-time series at a time, and while this has advantages, information on other time series or covariates can often be helpful for prediction. ARIMA is one such model, while VARIMA is an extension that considers multiple time series for forecasting. While the number of *E. coli* bloodstream infections per quarter would have also been a reasonable outcome to predict for our models, we wanted to apply the same method across all pathogens under mandatory surveillance. As the number of isolates tested for susceptibility per quarter for all other pathogens was considerably smaller (Supplementary Fig. 1), and given the short length of our time series, we did not consider these models.

While XGBoost was not designed with time series in mind, with the right feature engineering we could use it to address our problem, providing for each Trust–pathogen–antibiotic outcome, input features comprising the historical resistance for that Trust for all available pathogen–antibiotic combinations (not just the outcome), as well as all historical antibiotic usage rates for that Trust. One strength of our analysis is our in-depth domain knowledge: standardising usage for comparison between different Trusts, antibiotics and antibiotic formulations, and financial years, by using DDDs and number of occupied beds, rather than data-agnostic standardising methods; using resistance prevalences which are appropriate to answer questions about resistance in the population at risk. We also carefully considered the changes that we could expect to be able to predict from year to year in the context of the distribution of resistance prevalences, allowing us to demonstrate that XGBoost models indeed achieve better predictive performance in those Trusts where there were larger changes, without impairing performance in those where changes were smaller. Taking the difference between the previous two years forward was the worst performing model across all pathogen–antibiotic combinations, followed by LTF regression, indicating that some of the year-on-year observed differences may have been artefactual, related to the small numbers of isolates being tested and lack of representativeness of the population at risk. While LTF regression mitigated some of these fluctuations, it was either still influenced to a certain extent by the outliers and/or the linear model was not a good fit. For example, previous work found evidence for a sigmoid pattern in resistance trends, that is a fast rise following an initial period of low resistance levels, followed by a stable trend once a certain resistance percentage (below 100%) was reached[30]. However, for antibiotics that have been widely used for a long period of time, our period likely only covered the stable trend, not requiring a sigmoid. Given how little year-to-year variation there was, it was difficult for average measures of performance to massively outperform previous values taken forward, despite their ability to learn from previous resistance prevalences for all pathogen–antibiotic combinations, as well as previous antibiotic usage rates. Further, the hyperparameter tuning, feature selection, and feature engineering that we considered to improve generalisability did not improve performance (even though overfitting was reduced), with minimal decrease in mean absolute error in only some pathogen–antibiotic combinations and occasionally very small increases in some others. This is likely due to the reasonably small number of training examples which did not provide enough power to allow for learning of better hyperparameters than the default ones, which were set by the author of XGBoost based on empirical experimentation to work well on a diverse range of datasets.

One limitation of our study is the assumption that the bloodstream infections whose pathogens are tested for antibiotic susceptibility in each Trust are representative of the population being served by each Trust. For a high-income country, this may be reasonable given the severity of bloodstream infections means the vast majority of at-risk patients would have blood cultures taken, unlike low and middle-income countries where blood cultures are often only taken after empirical treatment failure[31]. It is still possible that some patients do not have cultures obtained or that prior antibiotics render cultures artificially negative, however, in our setting such patients would not be expected to have a different rate of antimicrobial resistance. Another limitation is the imperfect denominator for antibiotic usage as not everyone who occupied a day or overnight bed would have received antibiotics; however, this follows World Health Organisation recommendations[25], and makes features comparable both over time and between Trusts for our prediction models. Another limitation is the data aggregation to financial years, as previous studies have shown seasonality in the usage of many antibiotics and resistance in many pathogen–antibiotic combinations in the community[8,32]. Studies in the community rather than hospitals found the highest correlations with the antibiotics that were used most and that peaked during winter[32,33]. Unfortunately, the numbers were too small in our study for us to analyse the data quarterly across all pathogens considered; however, we did consider financial years to keep the winter months together. One alternative could have been to use smaller time periods and use both estimated resistance prevalence and some confidence limits on this (e.g. 90% CI) to represent uncertainty: however, we already had a large number of features for the number of observations. We also decided to predict resistance in FY2020-2021, despite this potentially being affected by COVID-19, given the relatively limited variation in usage over time (Supplementary Fig. 9). We only tried to predict resistance one year into the future due to our short time series. The short length of our time series is the main statistical limitation. In the future, as the length of the time series increases with more available data, deep learning architectures will become viable. Two good candidates are long short-term memory models, which can memorise both short and long-term information by learning to selectively remember or forget, and transformer models, which use the attention mechanism to learn which parts of the time series it should focus on. And as the time series grows even longer, predicting more than one year ahead may also become viable. Two possible approaches are to do this in an autoregressive manner by using the previously predicted year as the input for predicting the next year, or as an alternative, using encoder-decoder architectures to directly learn how to predict multiple steps ahead.

Antibiotic usage in the community has been shown to be associated with antimicrobial resistance. While some antibiotics we considered are not used in the community, others, e.g. amoxicillin, will be very common. The challenge is that incorporating these features into the model would require assigning community use to Trusts. The most recent English surveillance programme for antimicrobial utilisation and resistance reported associations between ethnicity and deprivation and higher AMR rates[34], suggesting these might be other potential predictive features. The association between antibiotic use and resistance is complex and is underlined by the One Health approach which aims to tackle AMR by approaching it from all compartments that are both affected, as well as contribute to this issue, namely human, animal and environmental[35].

The small variability we observed in resistance prevalence within Trust–pathogen–antibiotic combinations could be due to the (short) length of our time series, but it could reflect a plateau if resistance had already been increasing for quite a few years before our study[30], or resistance had become balanced with antibiotic usage in most Trusts, perhaps due to antibiotic stewardship practices[36]. One study used non-linear time series analysis to model relationships between antibiotic usage and resistance in five different populations in Europe for different pathogen–antibiotic combinations, as well as identify minimum usage

thresholds specific to each population to guide effective antimicrobial usage, balancing effectively treating the patient with preserving the effectiveness of antibiotics[37]. We note that both this study, as well as previous studies considering non-linear time-series analyses to identify antibiotic usage thresholds below which no further reduction in incidence of resistance was observed, considered much longer time periods than we, unfortunately, had available[38,39].

In summary, the change in resistance prevalence from year to year in a Trust–pathogen–antibiotic combination was generally small from FY2016-2017 onwards. However, focusing on those Trusts with larger changes, XGBoost, a machine learning model, provided better predictions of future resistance from historical antibiotic usage and historical resistance patterns in a variety of antibiotics and pathogens. Features with the highest overall contribution to predictions suggest that complex relationships were captured to achieve this performance. We therefore have a model that could be further tested and even deployed in a real-world setting to predict resistance prevalence in the next financial year, informing appropriate targeting of interventions and allocation of resources, in settings where notable changes in resistance prevalence take place.

## Data availability

National antibiotic resistance data, at a per hospital group (Trust) level, was obtained by aggregating data in the UKHSA SGSS, containing laboratory data supplied electronically by approximately 98% of hospital microbiology laboratories in England. A subset of the antibiotic resistance dataset is available through UKHSA's online data service, Fingertips[40]. Information on the use of antibiotics in secondary care was obtained from IQVIA (formerly QuintilesIMS, formed from the merger of IMS Health and Quintiles)[22]. All IQVIA data used retains IQVIA Solutions UK Limited and its affiliates' Copyright. All rights reserved. Use of IQVIA data for sales, marketing or any other commercial purposes is not permitted without IQVIA Solutions UK Limited's approval, expressed by IQVIA's Terms of Use. Datasets underlying the main figures are available as supplementary data (Supplementary Data 2–5 corresponding to Figs. 1–4).

## Code availability

Only open-source software was used for this analysis. R was used for the data preparation and visualisation of the results, while Python was used for modelling. Code for feature engineering and XGBoost models is publicly available at https://github.com/karinadorisvihta/AMR_forecasting[41].

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

## Acknowledgements

This work was supported by the National Institute for Health Research Health Protection Research Unit (NIHR HPRU) in Healthcare Associated Infections and Antimicrobial Resistance at Oxford University in partnership with the UK Health Security Agency (NIHR200915), and the NIHR Biomedical Research Centre, Oxford. D.W.E. is a Big Data Institute Robertson Fellow. A.S.W. is an NIHR Senior Investigator. D.A.C. was supported by the Pandemic Sciences Institute at the University of Oxford; the National Institute for Health Research (NIHR) Oxford Biomedical Research Centre (BRC); an NIHR Research Professorship; a Royal Academy of Engineering Research Chair and the InnoHK Hong Kong Centre for Cerebro-cardiovascular Engineering (COCHE). The views expressed are those of the authors and not necessarily those of the NHS, the NIHR, the Department of Health or the UK Health Security Agency. The funders/sponsors did not have any role in the design and conduct of the study; collection, management, analysis, and interpretation of the data; preparation, review, or approval of the manuscript; or decision to submit the manuscript for publication. For the purpose of Open Access, the authors have applied a Creative Commons Attribution CC BY public copyright license to any author-accepted manuscript version arising from this submission.

## Author contributions

K.-D.V., A.S.W., D.A.C. and D.W.E. designed the specific analysis. K.-D.V. conducted the analysis. K.-D.V., D.W.E. and A.S.W. draughted the manuscript. All authors, namely K.-D.V., E.P., K.B.P., S.H., R.L.G., K.H., D.C., R.H., B.M.-P., A.S.W., D.C. and D.W.E. contributed to interpretation of the results, revised and approved the manuscript for intellectual content, and had full access to all data analysis outputs (reports and tables) and take responsibility for their integrity and accuracy. K.-D.V. is the guarantor and accepts full responsibility for the work and conduct of the study, had access to the data, controlled the decision to publish, and attests that all listed authors meet authorship criteria and that no others meeting the criteria have been omitted.

## Competing interests

The authors declare the following competing interests: D.W.E. declares lecture fees from Gilead, outside the submitted work. Authors K.-D.V., E.P., K.B.P., S.H., R.L.G., K H., D.C., R.H., B.M.-P., A.S.W., D.C. declare no competing interests.

## Additional information

[1]Modernising Medical Microbiology, Experimental Medicine, Nuffield Department of Medicine, Level 7 Research Offices, John Radcliffe Hospital, Headley Way, University of Oxford, Oxford, UK. [2]The National Institute for Health Research Health Protection Research Unit in Healthcare Associated Infections and Antimicrobial Resistance, University of Oxford, Oxford, UK. [3]Department of Engineering Science, Institute of Biomedical Engineering, University of Oxford, Oxford, UK. [4]Health Economics Research Centre, Nuffield Department of Population Health, University of Oxford, Oxford, UK. [5]Healthcare-associated infections, Fungal, Antimicrobial resistance, Antimicrobial usage, & Sepsis Division, UK Health Security Agency, London, UK. [6]The National Institute for Health Research Oxford Biomedical Research Centre, University of Oxford, Oxford, UK. [7]OSCAR (Oxford Suzhou Centre for Advanced Research), University of Oxford, Suzhou, China. [8]Big Data Institute, Nuffield Department of Population Health, University of Oxford, Oxford, UK. [9]Department of Infectious Diseases and Microbiology, Oxford University Hospitals NHS Foundation Trust, John Radcliffe Hospital, Oxford, UK. [10]These authors contributed equally: Ann Sarah Walker, David Clifton, David W. Eyre.
✉e-mail: karina.vihta@gmail.com

