## [Peer Review File · Communications Medicine]

Reviewers' comments:

Reviewer #1 (Remarks to the Author):

This paper demonstrates the use of a machine-learning library, XGBoost, for predicting future AMR prevalence in hospital Trusts in UK using historical AMR levels.

The XGBoost results were compared to predictions obtained using simpler methods, namely: 1) previous value taken forwards, 2) difference between the previous two years taken, and 3) forwards and linear trend forecasting (LTF). The most basic and simple predictive model 1) achieved the lowest prediction errors, while XGBoost outperformed all other methods in Trusts with a larger change in AMR prevalence.

This manuscript has its merits, but I have important comments from the methodological point of view.

My interpretation of the main result is that when there is no change in the trend (AMR prevalence is constant in time for a certain Trust) a very naive model that states that future prevalence is the same as the present prevalence suffices. It is concerning that XGBoost was not able to predict such a simple pattern and I felt like XGBoost did not performed properly.

Can the author offer an explanation for this result? Initially I thought this is due to the typical uncertainty intrinsic to the ML models. If this is the case, even more care is needed in drawing the conclusions, as the differences between model 1 ("previous values taken forwards" and the XGboost errors are small and could be non-significant (within the same uncertainty range).

Also in figure 3, which hyperparameters were used in "XGBoost usage alone"?

In line 104, it is stated that "To our knowledge, machine learning methods have not been widely used for predicting resistance at an aggregate level such as a hospital, a network of hospitals, or a country". This is incorrect, in fact some manuscripts used XGBoost for AMR prediction, e.g., "Personal clinical history predicts antibiotic resistance of urinary tract infections." Nature Medicine 2019 and "Informing antimicrobial stewardship with explainable AI" PLOS Digit Health 2023. The authors should also carefully explain how their work differs from these already published manuscripts.

Line 183 states "XGBoost is not designed for time series, but with appropriate feature engineering and setup can be used for time series forecasting, especially as our time series are very short". The same concept is repeated in line 359. It appears that feature engineering is key, here. But apart from the code released by the authors, there are essentially no details, which need to be reported.

Minor comments:

In the abstract, it is stated that "Feature importance values indicated that complex relationships were exploited for predictions". In that form, it is an obvious statement (that is what any machine learning model does) and I'd recommend the authors to be more specific as in the explanation in line 315 "Previous resistance prevalence to the same antibiotic but in a different pathogen, as well as usage of the same antibiotic were also generally among the top 10 features, and similarly for other antibiotics from the same class"

line 147 "The Trust was obtained through linkage to mandatory surveillance data collected via the Healthcare-associated Infections Data Capture System" Maybe I am not reading it well, how can you obtain a Trust??

line 214, 2We chose to minimise the mean absolute error (mean of absolute difference between true and predicted value) as it is easily interpretable and less influenced by outliers than root mean squared error". I am not convinced this is a reason for using mean absolute errors, can the

author provide a reference? Aren't outliers important?

line 224 "To aid model interpretability, global feature importance was captured through mean absolute SHAP values, which measure the impact each feature has on the individual predictions, therefore higher values indicate more influential features". This sentence is not clear, please clarify if "higher values" refer to the Shap values or to their mean (and mean with respect to what?).

Reviewer #2 (Remarks to the Author):

Dear Authors,

This is immensely complex work and I would congratulate you on this. This is, to my knowledge, the first use of machine learning algorithms (MLA) and XG boost to step into the wider space of multiple pathogen/antibiotic susceptibility future modelling. These data will have broad appeal to clinicians and computer scientists alike.

My one over-arching comment relates to all of us in this field. It is very hard to follow the different mathematical approaches towards these problems and this creates fear/uncertainty in many readers. I am acutely aware of how this applies to my own work and not to be over-critical of this manuscript which is written well. I need needed to sometimes pause and re-read to understand (eg lines 153-56 made little sense, line 193 for training data during the COVID pandemic?). There can be some merit to explaining the methods to ensure the non-specialist reader can continue to follow.

Another comment relates to another common problem in that relatively few patients who receive antibiotics have matched microbiology data. Missing microbiology data are often because the blood cultures are collected too late after the initiation of antibiotics, and non uncommonly they are not collected at all. So we all have the problem of training and test data needing microbiological outcomes to drive predictive models, when this is not all blood stream infection cases. It does not mean there is an inherent problem with XGBoost or other methods for predicting AMR, but is possibly worth a mention.

Overall well done on this complex task and the very noteworthy observations made.

Reviewer #3 (Remarks to the Author):

The paper "Predicting future hospital antimicrobial resistance prevalence using machine learning," has been reviewed and is now addressing each point based on the paper's content:

The paper presents a novel machine learning approach, specifically utilizing the XGBoost model, to predict future antimicrobial resistance (AMR) at the hospital level. This claim is substantiated by a comprehensive comparison with traditional forecasting methods, demonstrating the model's superior predictive capability, particularly in scenarios of significant year-to-year AMR prevalence changes.

The novelty of applying the XGBoost model to predict AMR across multiple pathogens, antibiotics, and hospital groups is well articulated. The study's relevance to a broad audience, including public health policymakers and healthcare providers, is clear, emphasizing the global importance of combating AMR.

The research is positioned as an original contribution, building upon and distinguishing itself from existing studies through its aggregate level application of machine learning for AMR prediction. The paper effectively situates its findings within the current body of knowledge, referencing previous works to highlight its advancement beyond traditional correlation studies between antibiotic use

and resistance.

The argument for the XGBoost model's effectiveness in predicting AMR prevalence is convincingly presented with robust statistical analysis. However, the discussion on year-to-year resistance changes would benefit from additional exploration into external influencing factors, providing a deeper context for the observed trends.

This study is poised to significantly impact the field by showcasing the potential of machine learning in AMR management strategies. It suggests a paradigm shift towards more data-driven approaches in antimicrobial stewardship and infection control, highlighting the importance of integrating advanced data analysis methods.

The statistical approach, including the use of SHAP values for interpretability and the rationale behind selecting mean absolute error as a performance metric, is well-justified and executed. A discussion on potential statistical limitations or considerations of alternative analyses might enrich the validity of the findings.

Manuscript reference number: COMMSMED-23-0912

Title: Predicting future hospital antimicrobial resistance prevalence using machine learning

Note: Line and page numbers refer to the tracked manuscript with all revisions shown inline.

REVIEWER COMMENTS

Reviewer #1 (Remarks to the Author):

This paper demonstrates the use of a machine-learning library, XGBoost, for predicting future AMR prevalence in hospital Trusts in UK using historical AMR levels.

The XGBoost results were compared to predictions obtained using simpler methods, namely: 1) previous value taken forwards, 2) difference between the previous two years taken, and 3) forwards and linear trend forecasting (LTF). The most basic and simple predictive model 1) achieved the lowest prediction errors, while XGBoost outperformed all other methods in Trusts with a larger change in AMR prevalence.

This manuscript has its merits, but I have important comments from the methodological point of view.

1. My interpretation of the main result is that when there is no change in the trend (AMR prevalence is constant in time for a certain Trust) a very naive model that states that future prevalence is the same as the present prevalence suffices. It is concerning that XGBoost was not able to predict such a simple pattern and I felt like XGBoost did not performed properly.

Response: We would like to note that the most basic and simple model had mean absolute errors across all Trusts similar or slightly higher to those of the XGBoost models (Figure 3) and we have clarified this in the Results,

- both in the Abstract: “XGBoost models achieved the best predictive performance. Relatively limited year-to-year variability in AMR prevalence within Trust-pathogen-antibiotic combinations meant previous value taken forwards also achieved a low mean absolute error (MAE), similar to or slightly higher than XGBoost.” (lines 57-60), and
- the main text: “Overall, XGBoost models achieved the best predictive performance, with previous value taken forwards having very similar or slightly higher mean absolute error across all Trusts (Figure 3). The largest differences between previous value taken forwards and XGBoost were when XGBoost outperformed previous value taken forwards, eg for *P. aeruginosa* ceftazidime (2% difference, from 4% to 6%).” (lines 329-333).

Can the author offer an explanation for this result? Initially I thought this is due to the typical uncertainty intrinsic to the ML models. If this is the case, even more care is needed in drawing the conclusions, as the differences between model 1 ("previous values taken

forwards" and the XGboost errors are small and could be non-significant (within the same uncertainty range).

Response: When there is no change in trend, by definition, the naïve model of last observation carried forward will do very well, and we feel it is unrealistic for any machine learning model to do much better. This was the point that we tried to make by first introducing the performance of previous value taken forwards initially in our manuscript, which unfortunately led to the confusion, so we thank the reviewer for their comment and for helping us clarify the message. As above, we have clarified in the results both in the Abstract and the main text that XGBoost in fact does equally well or better across all Trusts.

2. Also in figure 3, which hyperparameters were used in "XGBoost usage alone"?

Response: We have clarified in the Methods that for these XGBoost models default hyperparameters were used: "For XGBoost models using historical antibiotic usage alone (no information on previous resistance prevalence) only models with default parameters were fitted." (lines 238-240). Have also added this detail to the Figure 3 legend, "with default parameters", and have also added the actual default parameters to Supplementary Methods: "Default XGBoost parameters as in main documentation <https://xgboost.readthedocs.io/en/stable/parameter.html>: eta=0.3, gamma=0, max_depth=6, min_child_weight=1, max_delta_step=0" (lines 17-19).

3. In line 104, it is stated that "To our knowledge, machine learning methods have not been widely used for predicting resistance at an aggregate level such as a hospital, a network of hospitals, or a country". This is incorrect, in fact some manuscripts used XGBoost for AMR prediction, e.g., "Personal clinical history predicts antibiotic resistance of urinary tract infections." Nature Medicine 2019 and "Informing antimicrobial stewardship with explainable AI" PLOS Digit Health 2023. The authors should also carefully explain how their work differs from these already published manuscripts.

Response: We have clarified in the Introduction that while ML methods such as XGBoost have been used in the past for AMR prediction **at an individual level** (using for example patient's personal clinical history as per the 2 example studies provided, added as references), they have not been used for predicting AMR **at an aggregate hospital level**: "To our knowledge, whilst machine learning methods have been used in the past for predicting resistance at an **individual** level, for example(11,12), they have not been widely used for predicting resistance at an **aggregate** level such as a hospital, a network of hospitals, or a country." (lines 108-111). We have also added the word aggregate in the Abstract Background for added clarity: "nationwide at an aggregate hospital level" (lines 44-45).

4. Line 183 states "XGBoost is not designed for time series, but with appropriate feature engineering and setup can be used for time series forecasting, especially as our time series are very short". The same concept is repeated in line 359. It appears that feature engineering is key, here. But apart from the code released by the authors, there are essentially no details, which need to be reported.

Response: We have re-ordered some of the phrases in the Methods to clarify that part of the engineering is in how the dataset is split into train and test sets in order for this to be appropriate for time series. We have also added details on how the time series lead to the creation of the input features and ultimately the input matrix:

- “XGBoost is not designed for time series, but with appropriate feature engineering and setup can be used for time series forecasting, especially as our time series are very short. Specifically, we used a training-test data split based on calendar time to train models and evaluate performance. We used percentage resistance in FY2020-2021 as our outcome for our training dataset” (lines 217-221).
- “All data available from the prior years of the time series was provided as input. Each pathogen-antibiotic-FY resistance prevalence and each antibiotic-FY usage rate with available data contributed a feature to the input matrix, with each Trust contributing data for one row. FY2021-2022 was used as our outcome for our test set.” (lines 223-226).

Minor comments:

5. In the abstract, it is stated that “Feature importance values indicated that complex relationships were exploited for predictions”. In that form, it is an obvious statement (that is what any machine learning model does) and I'd recommend the authors to be more specific as in the explanation in line 315 "Previous resistance prevalence to the same antibiotic but in a different pathogen, as well as usage of the same antibiotic were also generally among the top 10 features, and similarly for other antibiotics from the same class"

Response: We have expanded as suggested: “Feature importance values indicated that besides historical resistance to the same pathogen-antibiotic combination as the outcome, complex relationships between resistance in different pathogens to the same antibiotic/antibiotic class and usage were exploited for predictions. These were generally among the top 10 features ranked according to their mean absolute SHAP values. (lines 63-68).

6. line 147 “The Trust was obtained through linkage to mandatory surveillance data collected via the Healthcare-associated Infections Data Capture System” Maybe I am not reading it well, how can you obtain a Trust??

Response: We apologise, we have clarified that the Trust name matching the laboratory results had to be obtained through linkage as antimicrobial susceptibility results are reported by diagnostic laboratories, rather than by the Trusts:

"National antibiotic resistance data was obtained from the UK Health Security Agency's (UKHSA) Second Generation Surveillance System (SGSS), containing laboratory data supplied electronically by approximately 98% of hospital microbiology laboratories in England. Laboratory surveillance data were deterministically linked by UKHSA using date of birth, NHS number and specimen date to the Healthcare-associated Infections Data Capture System mandatory surveillance data to obtain hospital group (Trust) for each bacteraemia

case, following (19). We studied pathogens isolated from bloodstream infections subject to mandatory surveillance aggregated at Trust level" (lines 138-148).

The cases from each data source were linked together to identify NHS Trust level susceptibility testing data using standard UKHSA algorithms, as in the previous study ("Descriptive epidemiology of Escherichia coli bacteraemia in England, April 2012 to March 2014", Eurosurveillance, 2016), added as a reference.

7. line 214, 2We chose to minimise the mean absolute error (mean of absolute difference between true and predicted value) as it is easily interpretable and less influenced by outliers than root mean squared error". I am not convinced this is a reason for using mean absolute errors, can the author provide a reference? Aren't outliers important?

Response: Since for RMSE errors are squared before they are averaged, a relatively high weight is given to large errors, whereas for MAE this is not the case.

We wished to avoid over-influence from outliers as despite data cleaning, it is quite likely that some large outliers still arise from data quality issues. We are mostly concerned with optimising performance for most Trusts rather than those with large errors and hence choose to optimise MAE. We have clarified this in the text:

"We chose to minimise the mean absolute error (mean of absolute difference between true and predicted value) as it is easily interpretable and less influenced by outliers than root mean squared error, as for the former all errors are given the same weight, while for the latter more weight is given to larger errors. We wished to avoid over-influence from outliers as despite data cleaning, it is quite likely that some large outliers still arise from data quality issues. We are mostly concerned with optimising performance for most Trusts rather than those with large errors and hence choose to optimise mean absolute error." (lines 246-252).

8. line 224 "To aid model interpretability, global feature importance was captured through mean absolute SHAP values, which measure the impact each feature has on the individual predictions, therefore higher values indicate more influential features". This sentence is not clear, please clarify if "higher values" refer to the Shap values or to their mean (and mean with respect to what?).

Response: Have clarified: "To aid model interpretability, global feature importance was captured through mean absolute SHAP values calculated across all observations. SHAP values measure the impact each feature has on the individual predictions. These were computed on the test set for each Trust in each individual pathogen-antibiotic combination model. Large positive/negative values indicate features that have a significant impact on the prediction. Therefore, higher mean absolute SHAP values (calculated across all Trusts) indicate more influential features overall." (lines 262-267).

Reviewer #2 (Remarks to the Author):

Dear Authors,

This is immensely complex work and I would congratulate you on this. This is, to my knowledge, the first use of machine learning algorithms (MLA) and XG boost to step into the wider space of multiple pathogen/antibiotic susceptibility future modelling. These data will have broad appeal to clinicians and computer scientists alike.

1. My one over-arching comment relates to all of us in this field. It is very hard to follow the different mathematical approaches towards these problems and this creates fear/uncertainty in many readers. I am acutely aware of how this applies to my own work and not to be over-critical of this manuscript which is written well. I need needed to sometimes pause and re-read to understand (eg lines 153-56 made little sense, line 193 for training data during the COVID pandemic?). There can be some merit to explaining the methods to ensure the non-specialist reader can continue to follow.

Response: In terms of the specific sentences raised

- We have clarified: “Isolates tested are assumed representative of bloodstream infections in each Trust, as most detectable bloodstream infections are likely ascertained by widespread testing in serious illness in a high-income setting.” (lines 167-170).
- We have also clarified: “For some antibiotics susceptibility results were reported as susceptible or resistant. However, for a subset of the antibiotics, susceptibility results were split into susceptible, resistant, and a third intermediate category. Where susceptibility was reported as intermediate, this was considered susceptible in models following recommendations that this is susceptible under increased exposure(20)”. (lines 171-175).
- We have also expanded on the reasoning behind using FY2020-2021 as the outcome of our training dataset: “We used percentage resistance in FY2020-2021 as our outcome for our training dataset. Although this FY includes the start of the Covid-19 pandemic, this was the closest to the test set (outcome FY2021-2022), and maximised the history available for model training.” (lines 220-222).
- We have also added details on the exact calculation of usage rates: “Specifically, usage rates were calculated as the total DDDs per month divided by the number of days in the month, to obtain the mean DDDs per day, divided by mean number of day and overnight occupied beds per day and multiplied by 100. For example, an antibiotic usage of 40 DDDs amoxicillin per 100 bed-days, that is an usage rate of 40%, means 40% of inpatients receive one DDD of amoxicillin every day, an estimate of the therapeutic intensity.” (lines 183-189).
- The Methods have also been further clarified following the suggestions of the other 2 reviewers.

2. Another comment relates to another common problem in that relatively few patients who receive antibiotics have matched microbiology data. Missing microbiology data are often because the blood cultures are collected too late after the initiation of antibiotics, and non uncommonly they are not collected at all. So we all have the problem of training

and test data needing microbiological outcomes to drive predictive models, when this is not all blood stream infection cases. It does not mean there is an inherent problem with XGBoost or other methods for predicting AMR, but is possibly worth a mention.

Response: This was addressed as a limitation in our discussion which we have now expanded: "One limitation of our study is the assumption that the bloodstream infections whose pathogens are tested for antibiotic susceptibility in each Trust are representative of the population being served by each Trust. For a high-income country, this may be reasonable given the severity of bloodstream infections means the vast majority of at-risk patients would have blood cultures taken, unlike low and middle-income countries where blood cultures are often only taken after empirical treatment failure(30). It is still possible that some patients do not have cultures obtained or that prior antibiotics render cultures artificially negative, however in our setting such patients would not be expected to have a different rate of antimicrobial resistance." (lines 453-460). It was also stated as an assumption in previously lines 153-156, which has now been clarified as per suggestion above (lines 167-170).

Overall well done on this complex task and the very noteworthy observations made.

Response: Thank you!

Reviewer #3 (Remarks to the Author):

The paper "Predicting future hospital antimicrobial resistance prevalence using machine learning," has been reviewed and is now addressing each point based on the paper's content:

The paper presents a novel machine learning approach, specifically utilizing the XGBoost model, to predict future antimicrobial resistance (AMR) at the hospital level. This claim is substantiated by a comprehensive comparison with traditional forecasting methods, demonstrating the model's superior predictive capability, particularly in scenarios of significant year-to-year AMR prevalence changes.

The novelty of applying the XGBoost model to predict AMR across multiple pathogens, antibiotics, and hospital groups is well articulated. The study's relevance to a broad audience, including public health policymakers and healthcare providers, is clear, emphasizing the global importance of combating AMR.

The research is positioned as an original contribution, building upon and distinguishing itself from existing studies through its aggregate level application of machine learning for AMR prediction. The paper effectively situates its findings within the current body of knowledge, referencing previous works to highlight its advancement beyond traditional correlation studies between antibiotic use and resistance.

1. The argument for the XGBoost model's effectiveness in predicting AMR prevalence is convincingly presented with robust statistical analysis. However, the discussion on year-to-year resistance changes would benefit from additional exploration into external influencing factors, providing a deeper context for the observed trends.

Response: We have expanded our discussion on external influencing factors as advised: "Antibiotic usage in the community has also been shown to be associated with antimicrobial resistance(33). While some antibiotics we considered are not used in the community, others, e.g. amoxicillin, will be very common. The challenge is that incorporating these features into the model would require assigning community use to Trusts. The most recent English surveillance programme for antimicrobial utilisation and resistance reported associations between ethnicity and deprivation and higher AMR rates(34), suggesting these might be other potential predictive features. The association between antibiotic use and resistance is complex and is underlined by the One Health approach which aims to tackle AMR by approaching it from all compartments that are both affected as well as contribute to this issue, namely human, animal and environmental(35)." (lines 486-497).

This study is poised to significantly impact the field by showcasing the potential of machine learning in AMR management strategies. It suggests a paradigm shift towards more data-driven approaches in antimicrobial stewardship and infection control, highlighting the importance of integrating advanced data analysis methods.

Response: Thank you!

2. The statistical approach, including the use of SHAP values for interpretability and the rationale behind selecting mean absolute error as a performance metric, is well-justified and executed. A discussion on potential statistical limitations or considerations of alternative analyses might enrich the validity of the findings.

Response: The main statistical limitation is the length of the time-series. We have expanded on this in the discussion and have considered alternative machine learning models that should become viable in the future as the length of the time series increases: "The short length of our time series is the main statistical limitation. In the future, as the length of the time series increases with more available data, deep learning architectures will become viable. Two good candidates are long short-term memory models, which can memorize both short and long-term information by learning to selectively remember or forget, and transformer models, which use the attention mechanism to learn which parts of the time series it should focus on. And as the time series grow even longer, predicting more than one year ahead may also become viable. Two possible approaches are to do this in an autoregressive manner by using the previously predicted year as the input for predicting the next year, or as an alternative, using encoder-decoder architectures to directly learn how to predict multiple steps ahead." (lines 476-484).

REVIEWERS' COMMENTS:

Reviewer #1 (Remarks to the Author):

The authors satisfactorily addressed all comments from my previous report

Reviewer #3 (Remarks to the Author):

There are no more comments on revision.

No further changes were required as there were no further comments.

REVIEWERS' COMMENTS:

Reviewer #1 (Remarks to the Author):

The authors satisfactorily addressed all comments from my previous report

Reviewer #3 (Remarks to the Author):

There are no more comments on revision.